# Radiological and Physiological Predictors of IPF Mortality

**DOI:** 10.3390/medicina57101121

**Published:** 2021-10-18

**Authors:** Tomoo Kishaba, Akiko Maeda, Shoshin Yamazato, Daijiro Nabeya, Shin Yamashiro, Hiroaki Nagano

**Affiliations:** 1Department of Respiratory Medicine, Okinawa Chubu Hospital, Okinawa 904-2293, Japan; yambal917@gmail.com (S.Y.); yamayamashin@gmail.com (S.Y.); hiroakinoko322violin@gmail.com (H.N.); 2Department of Respiratory Medicine, Iizuka Hospital, Fukuoka 820-8505, Japan; orangepeco610@gmail.com; 3Department of Infectious, Respiratory, and Digestive Medicine, Graduate School of Medicine, University of the Ryukyus, Okinawa 903-0215, Japan; respiratoryyy@gmail.com

**Keywords:** IPF, anti-fibrotic agent, %FRC, soft tissue thickness, mortality

## Abstract

*Background and Objectives*: Idiopathic pulmonary fibrosis (IPF) has a variable clinical course, which ranges from being asymptomatic to progressive respiratory failure. The purpose of this study was to evaluate the novel clinical parameters of IPF patients who receive an anti-fibrotic agent. *Materials and Methods*: From January 2011 to January 2021, we identified 39 IPF patients at Okinawa Chubu Hospital. Clinical information was obtained, such as laboratory data, pulmonary function test (PFT) results, and chest images, including of soft tissue thickness and the high-resolution computed tomography (HRCT) pattern at diagnosis. *Results:* The mean age was 72.9 ± 7.0 (53–85); 27 patients were men and 12 were women. The mean body mass index was 25.1 ± 3.9 (17.3–35). Twenty-four were active smokers and the median number of packs per year was 20. Regarding laboratory findings, mean white blood cell (WBC), lactate dehydrogenase (LDH), and Krebs Von den Lungen-6 (KL-6) values were 7816 ± 1859, 248 ± 47, and 1615 ± 1503, respectively. In PFT, the mean percent predicted FVC, percent predicted total lung capacity, percent predicted functional residual capacity (FRC), and percent predicted diffusion capacity of the lung for carbon monoxide (DLco) were 66.8 ± 14.9%, 71.8 ± 13.7%, 65 ± 39.6%, and 64.6 ± 27.9%, respectively. In chest radiological findings, soft tissue thickness at the right 9th rib was 26.4 ± 8.8 mm. Regarding chest HRCT patterns, 15 showed the definite usual interstitial pneumonia (UIP) pattern, 16 showed the probable UIP pattern, and eight showed the indeterminate for UIP pattern. In the treatment, 24 patients received pirfenidone and 15 patients took nintedanib. The mean observation period was 38.6 ± 30.6 months and 24 patients died. The median survival time was 32.4 months (0.9–142.5). Multivariate analysis adjusted for age showed that both soft tissue thickness [Hazard ratio (HR): 0.912, 95% confidence interval (CI): 0.859–0.979, *p*-value: 0.009] and percent FRC [HR: 0.980, 95% CI: 0.967–0.992, *p*-value: 0.002] were robust predictors of IPF mortality. *Conclusions:* In IPF patients treated with anti-fibrotic agents, both soft tissue thickness at the right 9th rib shown on the chest radiograph and %FRC can be novel predictors of IPF mortality.

## 1. Introduction

Idiopathic pulmonary fibrosis (IPF) is a progressive and prevalent form of idiopathic interstitial pneumonia (IIP) [1]. IPF is difficult to diagnose due to a wide range of clinical presentations and differential diagnoses [2,3]. Until recently, no specific therapeutic drugs were available for IPF [4]. Prednisolone or immunosuppressants are commonly prescribed for IPF [5,6]. Anti-fibrotic agent such as pirfenidone or nintedanib have been introduced as newer therapeutic agents in clinical practice [7,8].

Several physiological [9,10] and radiological measures, such as forced vital capacity (FVC) diffusion capacity of the lung for carbon monoxide (DLco), traction bronchiectasis, and honeycombing have been reported as useful predictors of IPF mortality [11,12]. IPF has a variable clinical course, which ranges from asymptomatic to severe irreversible respiratory failure along with acute exacerbation [13]. The prediction of the clinical course is crucial for chest physicians. The aim of this study was to identify radiological and physiological predictors of IPF mortality.

## 2. Methods

### 2.1. Study Population and Collection Data

This research comprised a retrospective study, which focused on a chart review of medical records. Therefore, our institutional review board waived informed consent for each patient.

From January 2011 to January 2021, ninety-six IPF patients were diagnosed at Okinawa Chubu Hospital. Thirty-two patients received prednisolone alone or a combination of anti-fibrotic agents and immunosuppressants. Twenty-five patients were followed-up without treatment because of clinical stability. Thirty-nine IPF patients received an anti-fibrotic agent, such as pirfenidone or nintedanib, during the observation period. Clinical information was gathered, including age, gender, smoking history, body mass index (BMI), dyspnea, modified medical research council (mMRC) dyspnea score [14], and cough and symptom duration at diagnosis of IPF. BMI was followed for one year. The serum white blood cell (WBC), lactate dehydrogenase (LDH), and Krebs Von den Lungen-6 (KL-6) were collected.

### 2.2. Physiological Data

FVC, percent predicted FVC (%FVC), total lung capacity (TLC), percent predicted TLC (%TLC), functional residual capacity (FRC), percent predicted FRC (%FRC), and percent predicted DLco (%DLco) were evaluated. FRC was calculated by the gas dilution method with helium. DLco was measured with the single-breath method. In addition, we also evaluated composite physiological index (CPI) [15], and gender-age-physiology (GAP) score [16]. IPF severity was evaluated by GAP score.

### 2.3. Chest Imaging Information

The soft tissue thickness from the chest radiograph of the posterior-anterior view in an erect position was assessed. The positive associations between BMI and progression of IPF were previously described in the literature [17,18]. The soft tissue thickness of the right 9th rib is usually the thinnest in the thoracic cage [19]. The right 9th rib is an adequate anatomical landmark for the evaluation of soft tissue thickness. The measurement of the soft tissue thickness at the right 9th rib is outlined in Figure 1. The distance between the outer edge of soft tissue and that of the right 9th rib was defined as soft tissue thickness measured on the posterior-anterior view. Furthermore, we reviewed the chest high-resolution computed tomography (HRCT) pattern at diagnosis of IPF based on the latest international IPF guideline [20].

### 2.4. Survival Period

The diagnosis date was set as the date of pulmonary function test (PFT) for each patient. The follow-up period ended in August 2021. The survival time was defined from the PFT date until the date of death, or until August 2021 if the patient was alive at that time. Among the causes of death, acute exacerbation of IPF was defined as idiopathic or triggered development of respiratory failure with bilateral new shadow superimposed fibrosis within one month based on the criteria of an international project [21].

### 2.5. Data Analysis

Each clinical measure is expressed as the mean or median based on a normal distribution where appropriate, and standard deviation, minimum and maximum are shown.

Categorial data were expressed as a number. For evaluation of correlation, we derived the Pearson product moment. Univariate and multivariate analyses with the Cox proportional hazards model were performed. The hazard ratio, 95% confidence interval (CI), and *p*-value were calculated and analyzed. The adequate threshold of predictors of IPF mortality were analyzed by the receiver operating characteristic (ROC) curve. The Kaplan–Meier survival curve with the log-rank test was used for survival measurement.

## 3. Results

### 3.1. Baseline Description

The medical information of 39 IPF patients who received anti-fibrotic agents were collected over 10 years, as shown in Table 1. The mean age was 72.9 ± 7.0; 27 were men and 12 were women. Twenty-four were current or ex-smokers. The mean BMI was 25.1 ± 3.9. In terms of clinical symptoms, the mean MRC was 1.3 ± 0.8 and 30 patients noted non-productive cough. The mean disease duration was 36.4 ± 30.7 months.

In laboratory findings, mean serum WBC, LDH, and KL-6 were 7816 ± 1859/ μL, 248 ± 47 U/L, and 1615 ± 1503 U/mL, respectively. In radiological findings, mean soft tissue thickness at the right 9th rib was 26.4 ± 8.8 mm. Regarding chest HRCT pattern, 15 definite UIP patterns,16 probable UIP patterns, and eight indeterminate for UIP patterns were found. All indeterminate for UIP pattern patients underwent video-associated thoracoscopic surgery and arrived at a final diagnosis of IPF through a multi-disciplinary discussion. The mean observation period was 38.6 ± 30.6 months.

### 3.2. Pulmonary Function Test

In terms of PFT, mean FVC, %FVC, TLC, %TLC, FRC, %FRC, and %DLco were 1.92 ± 0.56 L, 66.8 ± 14.9%, 3.30 ± 0.74 L, 71.8 ± 13.7%, 1.56 ± 0.85 L, 65.0 ± 39.6%, and 64.6 ± 27.9, respectively, as shown in Table 2. In the composite index of PFT, mean CPI was 59.4 ± 26.7 and mean GAP score was 4.9 ± 1.8. Regarding classification of GAP stage, nine patients were classified as stage I, 15 were classified as stage II, and 15 were classified as stage III.

In treatment, twenty patients received pirfenidone continuously and four patients switched from pirfenidone to nintedanib. Fifteen patients received nintedanib continuously.

Among 39 patients, 24 patients died during the observation period. Causes of death were as follows: six died due to acute exacerbation, 10 died due to progression of IPF, four died due to pneumonia, three died due to lung cancer, and one died due to diffuse alveolar hemorrhage, as shown in Figure 2.

### 3.3. Hazard Analyses for IPF Mortality

To identify associations between clinical parameters and IPF mortality, univariate and multivariate analyses were performed for the IPF cohort using the Cox proportional hazards model, as shown in Table 3. Parameters considered were age, BMI, serum LDH, KL-6, soft tissue thickness, FVC, %FVC, TLC, %TLC, FRC, %FRC, and %DLco. Soft tissue thickness, FRC, and %FRC showed statistical significance. After adjusting for age, both soft tissue thickness and %FRC were predictors of IPF mortality. Regarding treatment, both pirfenidone and nintedanib showed no statistically significant prediction of IPF mortality. The Pearson product moment of GAP and soft tissue thickness was found to be −0.532, whereas that of GAP and %FRC was found to be −0.529. We also estimated survival based on gender. The mean survival of men and women was 38.1 ± 29.2 months and that of women was 39.7 ± 35.0 months. The *p*-value was 0.558, which was not statistically significant.

### 3.4. Survival Curve Based on Predictors of IPF Mortality

The ROC curve showed the adequate threshold of IPF mortality was a soft tissue thickness of about 26 mm. The area under the curve of 26 mm was 0.658 (Figure 3). The Kaplan–Meier survival curve in the under 26 mm group showed poor prognosis compared to the over 26 mm group (*p* = 0.01) (Figure 4).

Another ROC analysis showed the threshold of IPF mortality was 65% in %FRC. The area under the curve of 65% was 0.55 (Figure 5). The Kaplan–Meier survival curve indicated the under 65% group showed a poor prognosis compared to the over 65% group (*p* < 0.01) (Figure 6).

## 4. Discussion

In this retrospective study, both soft tissue thickness and %FRC were identified as predictors of IPF mortality in this cohort. The physiological and radiological parameters such as FVC, DLco, traction bronchiectasis, and honeycombing are routinely used [22,23]. The chest radiograph is easy to use and cost effective in clinical practice, as an alternative to HRCT, and provides useful new information for clinicians. Regarding the role of the chest radiograph for IPF patients, both distribution of fibrosis and volume loss of the bilateral lower lung field have been addressed [24,25]. Chest HRCT has played a major role in the diagnosis and treatment response of IPF patients [26,27,28]. However, performing CT scans is costly and involves excessive exposure to radiation [29]. The search for cheaper and easier means to predict IPF mortality in patients in daily clinical practice has therefore been considered. The assessment of soft tissue thickness at the right 9th rib provides a new approach to evaluate IPF patients. In addition, the soft tissue in the thorax may have associations with nutrition and disease progression [30]. The delta BMI predicted IPF prognosis in this cohort [17]. Malnutrition and reduced BMI are associated with a poor prognosis [31,32]. The relationship between soft tissue thickness and delta BMI or nutritional status can be another important issue for IPF patients.

Mortality prediction by %FRC in IPF patients is a novel finding of our study. Pathological and radiological findings have been considered to play a major role in IPF evaluation [33,34,35] because IIPs were originally developed as a disease entity from a pathological perspective [36]. In the algorithm of IPF diagnosis, the chest HRCT plays a central role in diagnosis [37], and the international guidelines for IPF have insisted on the importance of the interpretation of the chest HRCT pattern [20]. The weights assigned to HRCT and pathology reflect their value; however, physiological evaluation has also been proposed using PFT parameters such as FVC and DLco as surrogate markers of IPF mortality [38,39]. In terms of DLco, baseline %DLco is a useful predictor of fibrotic ILD [40]. Furthermore, %DLco and pathological fibrotic foci were robust predictors of acute exacerbation of IPF patients [41]. In FVC, many clinical studies have shown that baseline FVC or a decline in FVC predicts IPF progression or survival [42,43]. On the contrary, FRC was not sufficiently evaluated for IPF patients until now. In PFT, FRC is sometimes associated with total lung capacity or FVC [44,45]. IPF is a parenchymal restrictive disorder. Therefore, our result of %FRC prediction of IPF mortality provides a new and significant consideration for reviewing full PFT in IPF patients. In addition, DLco cannot be performed in the case of reduced vital capacity. However, FRC can be undertaken for patients having some degree of reduced PFT. The easy application of FRC rather than DLco or surgical biopsy has clinical implications for chest physicians.

## 5. Limitations and Strengths

This study has several limitations. First, this was a single-center retrospective study. Therefore, the results will not be applicable to the entire global IPF population. Nonetheless, the age and gender balance of our cohort was similar to that of patients of previous research [46]. Second, our study population was small. However, we carefully chose IPF patients who continuously received anti-fibrotic agents. Thus, clinical data, such as respiratory symptoms, laboratory biomarkers, and physiological and radiological findings, may not be biased to a large extent compared to previous IPF cohorts [47]. Third, in our evaluation of soft tissue thickness, we were unable to collect detailed nutritional data, such as that relating to serum albumin, lymphocyte, or transferrin. However, we consistently evaluated soft tissue thickness in the posterior-anterior view in the same location and in the same manner. Fourth, we could not compare soft tissue thickness or %FRC with chest HRCT findings or serum biomarkers. These issues may be considered future topics for IPF patients.

The strength of our study was the comprehensive collection of data relating to patients’ characteristics, including clinical symptoms, detailed PFT parameters, radiological evaluation, and treatment information.

## 6. Conclusions

Soft tissue thickness at the right 9th rib and %FRC may have a role in the prediction of IPF mortality. The chest radiograph and PFT is easily available and can be performed in every institution. Future prospective multi-center studies can help to confirm our findings.

## Figures and Tables

**Figure 1 medicina-57-01121-f001:**
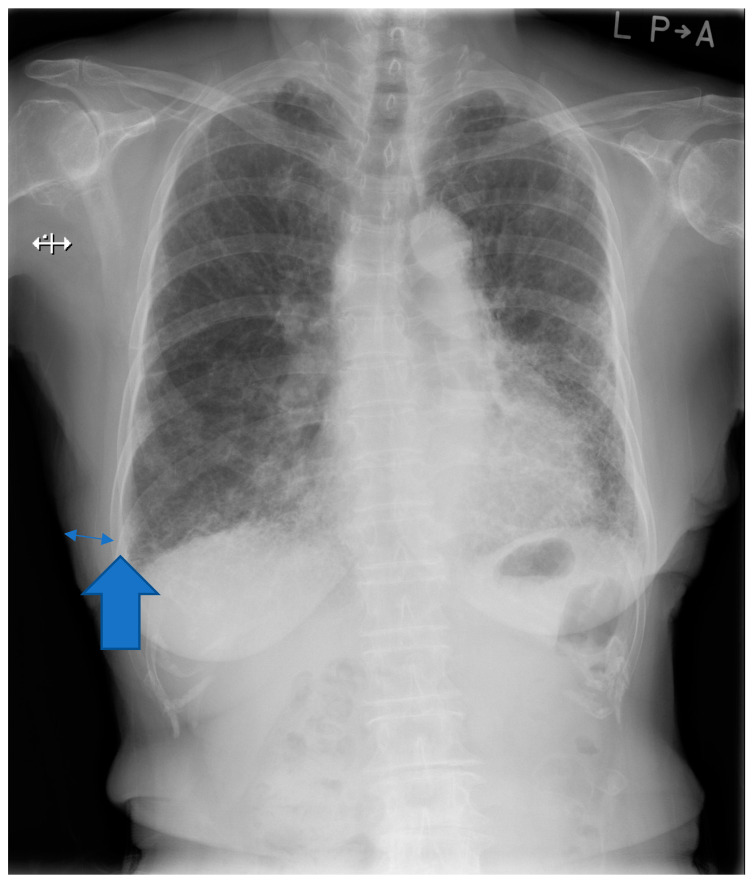
Evaluation of soft tissue thickness in the chest radiograph. Thin double arrow indicates soft tissue thickness. Thick arrow indicates outer edge of the right 9th rib.

**Figure 2 medicina-57-01121-f002:**
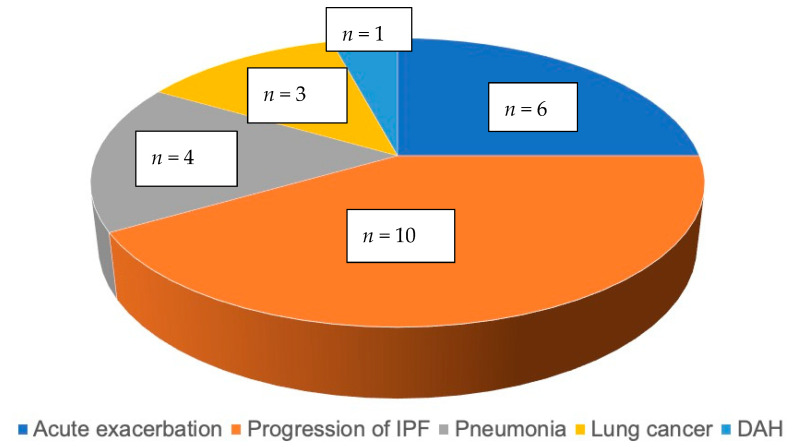
Causes of death (*n* = 24). Abbreviations: IPF; Idiopathic pulmonary fibrosis, DAH; diffuse alveolar hemorrhage.

**Figure 3 medicina-57-01121-f003:**
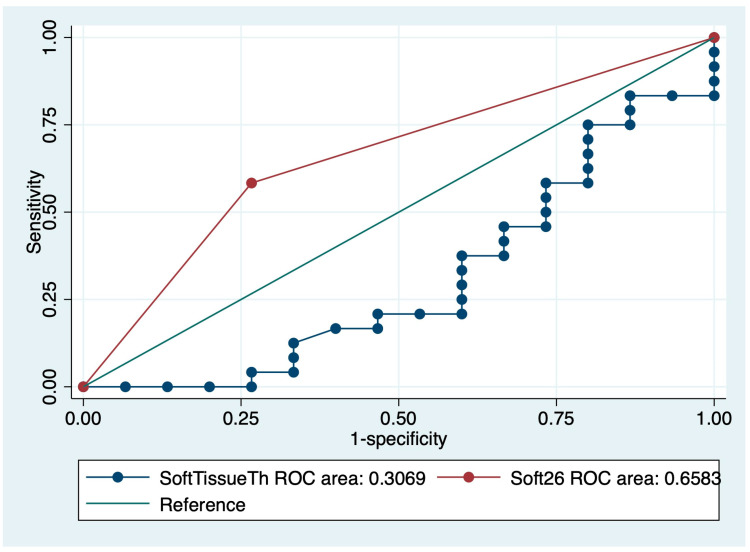
ROC curve of soft tissue thickness for IPF mortality.

**Figure 4 medicina-57-01121-f004:**
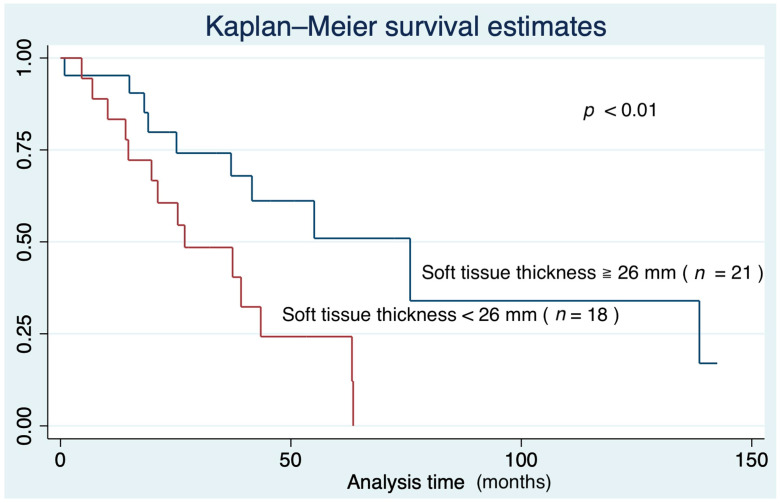
Kaplan–Meier survival curve based on soft tissue thickness.

**Figure 5 medicina-57-01121-f005:**
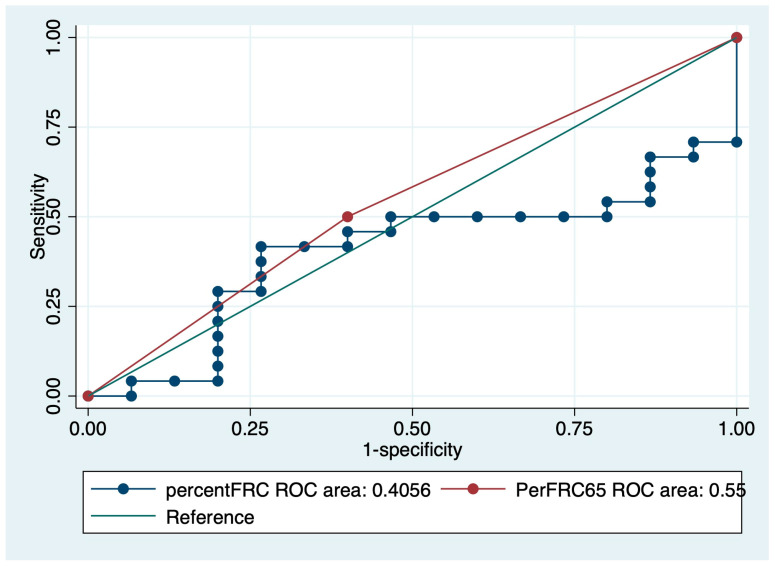
ROC curve of %FRC for IPF mortality.

**Figure 6 medicina-57-01121-f006:**
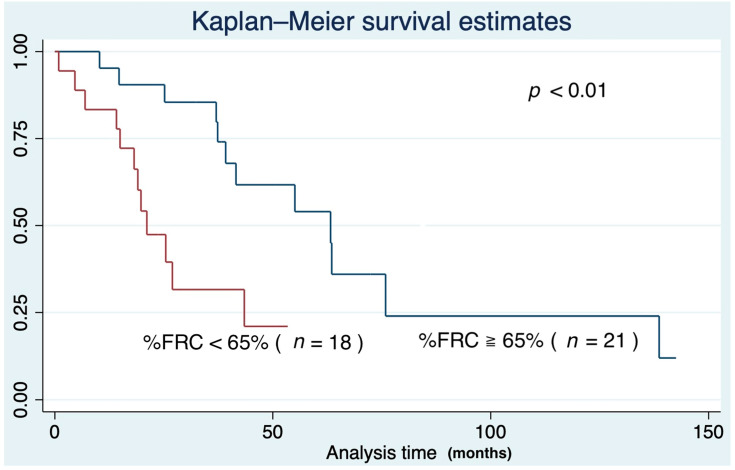
Kaplan–Meier survival curve according to the %functional residual capacity.

**Table 1 medicina-57-01121-t001:** Clinical characteristics of IPF patients receiving anti-fibrotic agent (*n* = 39).

Age	72.9 ± 7.0 (53–85)
Gender (Men/Women)	(27/12)
Smoker (active/never)	(24/15)
Pack-year	20 (0–100)
BMI (kg/mm^2^)	25.1 ± 3.9 (17.3–35)
mMRC	1.3 ± 0.8(0–3)
Cough (Yes/No)	(30/9)
Disease duration (months)	36.4 ± 30.7(2–120)
WBC (μL)	7816 ± 1859 (4700–11700)
LDH (U/L)	248 ± 47 (173–398)
KL-6 (U/mL)	1615 ± 1503 (332–6682)
Soft tissue thickness (mm)	26.4 ± 8.8 (1.7–52.2)
HRCT patten (definite/probable /indeterminate)	(15/16/8)
Observation period (months)	38.6 ± 30.6 (0.9–142.5)
Survival time (months)	32.4 (0.9–142.5)

Data are expressed as mean ± standard deviation, minimum and maximum, or median. Abbreviations: BMI; Body mass index, mMRC: modified Medical Research Council dyspnea score, WBC; white blood cell, LDH; lactate dehydrogenase, KL-6; Krebs Von den Lungen-6, HRCT; high-resolution computed tomography.

**Table 2 medicina-57-01121-t002:** Physiological data of IPF patients (*n* = 39).

FVC (L)	1.92 ± 0.56 (0.85–3.26)
Precent predicted FVC (%)	66.8 ± 14.9 (31.5–94.8)
TLC (L)	3.30 ± 0.74 (2.06–5.01)
Precent predicted TLC (%)	71.8 ± 13.7 (45.1–115.7)
FRC (L)	1.56 ± 0.85 (0.71–2.92)
Precent predicted FRC (%)	65.0 ± 39.6 (25.4–143.1)
Precent predicted DLco (%)	64.6 ± 27.9 (28.1–136.1)
Composite physiological index	59.4 ± 26.7 (0.3–104)
GAP score	4.9 ± 1.8 (1–8)
GAP stage (I/II/III)	(9/15/15)

Data are expressed as mean ± standard deviation, minimum and maximum, or median. Abbreviations: FVC; forced vital capacity, TLC; total lung capacity, FRC; functional residual capacity, DLco; diffusion capacity of the lung for carbon monoxide, GAP; Gender-Age-Physiology.

**Table 3 medicina-57-01121-t003:** Results of hazards analyses for IPF mortality (*n* = 39).

	Hazard Ratio	95%CI	*p*-Value
Univariate			
Age at PFT, years	1.07	1.01–1.12	0.02
BMI (kg/mm^2^)	0.89	0.79–1.00	0.06
LDH (U/L)	1.00	0.99–1.01	0.78
KL-6 (U/mL)	0.99	0.99–1.00	0.37
Soft tissue thickness (mm)	0.90	0.85–0.96	<0.01
FVC	0.74	0.37–1.46	0.38
%FVC	0.99	0.97–1.01	0.44
TLC	0.66	0.24–0.33	0.24
%TLC	0.99	0.96–1.02	0.44
FRC	0.38	0.22–0.67	<0.01
%FRC	0.98	0.97–0.99	<0.01
%DLco	0.99	0.96–1.01	0.34
Pirfenidone	0.88	0.76–1.03	0.12
Nintedanib	0.75	0.32–1.77	0.51
Multivariate			
BMI (kg/mm^2^)	0.91	0.80–1.03	0.13
Soft tissue thickness (mm)	0.92	0.86–0.98	<0.01
FRC	0.41	0.23–0.72	<0.01
%FRC	0.98	0.97–0.99	<0.01

Abbreviations: CI; confidence interval, BMI; body mass index, LDH; lactate dehydrogenase, KL-6; Krebs Von den Lungen-6, FVC; forced vital capacity, TLC; total lung capacity, FRC; functional residual capacity, DLco; diffusion capacity of the lung for carbon monoxide.

## Data Availability

Data sharing not applicable.

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
