# Peer review of "Radiological and Physiological Predictors of IPF Mortality"

_medicina, 2021, doi:10.3390/medicina57101121_

Round 1

Reviewer 1 Report

To Editor,

This single retrospective study by Kishaba et al identified novel clinical parameters of IPF patients who received anti-fibrotic drugs. They evaluated the basic laboratory findings, pulmonary function test and chest high resolution CT. 24 patients received pirfenidone and 15 patients received nintedanib. the results indicated that soft tissue thickness and FRC% were robust predictors of IPF mortality. The finally conclude that soft tissue thickness at 9th rib on the chest radiograph and %FRC were novel predictors of IPF mortality in IPF patients receiving anti-fibrotic drugs.

The manuscript is well written, however I have some concerns. The manuscript should be proofread by a native English speaker. Throughout the manuscript, there are couple of typos, grammatical error and spelling mistakes.

a) the "n" number is very low. Therefore, to predict that soft tissue thickness and %FRC as novel predictors should be made cautiously.

b) In Fig 1, authors should indicate what exactly they are measuring and figure legend should be described in detail

c) How the severity of IPF was determined? Criteria should be mentioned in the methods.

d) Does the soft tissue thickness and %FRC correlate with severity of IPF?

Reviewer 2 Report

There was a large difference in the percent of men and women in the study. Where there any gender-based differences in outcomes?

Figure 1 is too big and not enough detail is given in the caption. What is the significance of the giant blue arrow?

Figure 2 should have some quantitative metrics and include more description in the caption. 

The grey background behind figure 3 and 4 is distracting and not needed. Please remove it. 

Reviewer 3 Report

Idiopathic pulmonary fibrosis is a serious condition of the lung that is not rare at all but is not understood well in terms of etiology and prognosis. Therefore, the manuscript by Kishaba et al. is generally a very welcome contribution to this topic. The applied statistical methods appear to be suitable. However, there are several points that need to be improved, before a publication can be recommended.

Before addressing the content of the manuscript, a statement on the English writing has to be made. We are aware that for scientists from Japan it is especially challenging to write text in English. This is visible also in this manuscript. Regardless of the content, the text of the whole manuscript has to be improved as there are many grammatical mistakes. We do not see it as a task of the reviewer to provide a full correction here. Therefore, only some examples are listed in the section of minor points below. Generally, I would recommend to call in a native speaker to check the manuscript or the use of an according (online) service.

Major points:

The number of observed participants in this study is rather low (36) and the general relevance is further limited by the fact that the patients of this study are already on a specific treatment. The treatment was apparently not the same for all observed patients, but this differences have apparently not been considered/analyzed/discussed as a possible determinant of mortality or even as clinical characteristics. Age, BMI and smoking status have been considered, but not the type of treatment.

Hazard rates for soft tissue thickness and %FRC are close to one. They might appear as clear "predictors" in the whole survival analysis, but the identified parameters account only for a small part of the mortality rate. Therefore the use as predictors in individual diagnosis is rather limited and the term "robust predictor" appears not really justified.

As survival time studies are very sensitive to temporal changes in parameters, it is vital to show if factors have changed during the observation period. It is not clear from the manuscript, if factors like smoking status or BMI have been followed for or were just collected at the start of the observation. The same is true for pharmaceutical treatment.

Figure 1: The image provided does not really show how a soft tissue thickness measurement has been performed. The meaning of the blue arrow is unclear. The indicator in the radiograph seems not to show any visual limits that could be interpreted as soft tissue thickness.

Minor points:

The authors used a ROC-curve to determine threshold values for specific parameters. As ROC curves are valuable indicators of model quality, the curves (f.e. in a supplement) or at least according ROC-AOC values (in table) should be provided.

Line 20: Abbreviations (FRC) should be explained at the first point of use.

Grammar and spelling (examples):

Line 35, 44: "IPF have many..."

Line 46: "Based on the previous study..."

Line 52: "...consists..."

Line 60: "This study is retrospective study..."

Line 83: "When patient died..."

Line 84: "If patient survive..."

Line 92: Wording: "We judged p<0.05 as statistically significant"

Line 115-117: Sentence is missing a verb..

Line 121, tables: Data are expresses...

Line 124: "patiens"

Line 126, Fig 2: "Causes of death were 6 acute exacerbation." Define what exactly was exacerbated!

Line 135-137: "Among age... showed statistical significance."

Line 144: "..mortality was 26 mm about soft tissue..."

Line 156-157: Sentence is missing verb.

Line 162: "performing CT scan is high cost"

Line 176: "We acknowledge the weight of HRCT and pathology."

Round 2

Reviewer 1 Report

The manuscript is significantly improved and authors addressed all my concerns. The authors can elaborate the figure legends.

Fig 4, tissue spelling is incorrect.